# Cellular Metabolic Disorders in a Cohort of Patients with Sjogren’s Disease

**DOI:** 10.3390/ijms26104668

**Published:** 2025-05-13

**Authors:** Julian L. Ambrus, Alexander Jacob, Abhay A. Shukla

**Affiliations:** 1Department of Medicine, SUNY at Buffalo School of Medicine, 875 Ellicott Street, Buffalo, NY 14203, USA; ajacob6@buffalo.edu; 2Immco Diagnostics of Trinity Biotech, Amherst, NY 14228, USA; abhay.shukla@trinitybiotech.com

**Keywords:** Sjogren’s disease, metabolism, autoantibodies

## Abstract

Metabolism disorders have been seen in multiple autoimmune diseases, including SLE and Sjogren’s disease. The current studies were designed to evaluate mutations in genes involved in metabolism in a cohort of patients with Sjogren’s disease, diagnosed from clinical criteria and the presence of antibodies to salivary gland antigens. Patients were from an Immunology clinic that follows a large population of patients with autoimmune and metabolic disorders. The patients included in these studies were patients who met the criteria for Sjogren’s disease and for whom we were able to obtain genetic studies, sequencing of the mitochondrial DNA, and whole exome sequencing. There were 194 of these patients, and 192 had mutations in one or more gene involved in metabolism: 188 patients had mutations in mitochondrial respiratory chain genes, 17 patients had mutations in mitochondrial tRNA genes, 10 patients had mutations in mitochondrial DLOOP regions, 6 patients had mutations involved in carnitine transport, 6 patients had mutations in genes causing mitochondrial depletion, and 7 patients had glycogen storage diseases. In all cases, the treatment of the metabolic disorder led to symptomatic improvement in energy, exercise tolerance, gastrointestinal dysmotility, and the management of infections. In conclusion, metabolic disorders are common in patients with Sjogren’s disease and may be one of the factors leading to the initiation of the disease. The treatment of patients with Sjogren’s disease should include the treatment of the underlying/associated metabolic disorder.

## 1. Introduction

Our understanding of the pathophysiology of autoimmune diseases has been rapidly advancing over the last several years. While dysregulated immune function has been appreciated for decades, the appreciation of dysregulated metabolism in autoimmune diseases is relatively recent [1,2,3,4,5,6,7]. Abnormal mitochondrial function was first observed in SLE [8,9,10,11,12,13,14,15] but has been observed in Sjogren’s disease as well [16,17,18,19,20,21,22,23,24,25,26]. The analysis of metabolic disorders in the innate and adaptive immune cells of 30 patients with Sjogren’s disease documented alterations in multiple genes involved in mitochondrial metabolic pathways, along with histologic abnormalities in mitochondria [17]. Abnormal mitochondria in the salivary glands of patients with Sjogren’s disease have been observed in other studies as well [27,28]. One study suggested that the abnormal production of cytochrome c in the salivary glands of patients with Sjogren’s disease contributed to the apoptosis of salivary gland tissues [29]. No previous studies have examined genes involved in systemic metabolic disorders in patients with Sjogren’s disease that could theoretically contribute to the development and/or manifestations of the disease. Fatigue, exercise intolerance, gastrointestinal dysmotility, accelerated osteoarthritis, and difficulty handling infections is frequently seen in patients with adult-onset metabolic diseases [26,30,31,32]. Many patients with Sjogren’s disease have similar symptoms [33]. We previously observed, in a limited numbers of patients, that many of the symptoms attributable to Sjogren’s disease were in fact related to the underlying metabolic disease [24]. We therefore sought to determine the number of patients with mutations in genes associated with metabolism in a cohort of patients with Sjogren’s disease seen in our clinics.

## 2. Materials and Methods

### 2.1. Patients

All the patients discussed in this manuscript were followed at the Immunology clinics of SUNY at the Buffalo School of Medicine. They had clinical symptoms of xerostomia and xerophthalmia with positive Schirmer’s tests performed by their Ophthalmologists. Some patients demonstrated decreased salivary flow. All patients underwent serologic testing for antibodies to SSA, SSB, salivary gland protein 1 (SP1), carbonic anhydrase 6 (CA6), and parotid secretory protein (PSP) as part of their routine medical care. Because all patients had complaints of fatigue, exercise intolerance, accelerated osteoarthritis, and recurrent infections, genetic studies were obtained as part of their routine medical care. Twenty-seven percent of the patients had associated gastrointestinal dysmotility. The patients ranged in age from 21 to 81 years (mean 54.2 +/− 13.5 years). Eighty-nine percent of the patients were female.

### 2.2. Genetic Studies

Sequencing of the mitochondrial genome and whole exome sequencing were performed by GeneDx (Gaithersburg, MD, USA).

## 3. Results

The first issue to address is the autoantibodies expressed by the patients in this study. They all met the American–European clinical criteria for Sjogren’s disease with clinical signs of dry eye and dry mouth and positive Schirmer’s tests, except only 13 of the patients had SSA antibodies. All the patients had autoantibodies associated with Sjogren’s disease, but the majority had antibodies to SP1 and CA6 (Figure 1), which are salivary gland-specific antigens [34,35,36,37]. Many of the patients expressed more than one autoantibody. Interestingly, 54% of the patients with SP1 autoantibodies expressed IgM autoantibodies, while 67% of the patients with CA6 autoantibodies expressed IgG autoantibodies. These patients might by called seronegative Sjogren’s patients by some investigators because of their lack of SSA expression.

Genetic studies looking for metabolic disorders were conducted on these patients because of symptoms that are consistent with adult-onset metabolic disorders: fatigue, exercise intolerance, recurrent infections, accelerated osteoarthritis, gastrointestinal dysmotility, including gastroparesis, gastroesophageal reflux and constipation, and, in some cases, dyspnea. In all cases, whole exome sequencing and sequencing of the mitochondrial genome were performed by GeneDx. Of the patients studied, 192 (99%) had mutations in genes associated with metabolic function (Figure 2). Some patients carried more than one mutation. Rare missense mutations in mitochondrial respiratory chain genes were common: complex 1–76, complex 3–44, complex 4–19, and complex 5–49. One patient had a mutation in the succinate dehydrogenase gene, which is involved in complex 2 of the mitochondrial respiratory chain but also the citric acid cycle. Six patients had mutations associated with carnitine: CPT2—4 and SLC22A5—2. Mutations in various mitochondrial tRNA were seen in 17 patients, and 10 patients had rare MT-DLOOP mutations. One patient had a PDSS2 mutation associated with CoQ10 deficiency, and six patients had mutations associated with mitochondrial depletion syndrome: POLG—three; MCME1—one; RRMP8—one; and thymidine kinase—one. Seven patients had mutations in genes causing glycogen storage diseases: Pompe disease—one; Forbes–Cori disease—one; McArdle’s Disease -1, phosphofructokinase deficiency (type IX)—1, and lactate dehydrogenase deficiency (type XI)—three. In short, mutations in mitochondrial respiratory chain genes were most common (189), other mutations affecting mitochondrial function were less common [38], and mutations in enzymes involved in glycogen storage pathways were least common [7]. The result of this in all these mutations is inefficient generation of ATP, and in the case of glycogen storage diseases, difficulty handling complex carbohydrates.

All patients received treatment for their underlying metabolic disorder. The treatment of mitochondrial disorders involves several medications, each of which works through a different mechanism, so a synergistic effect is seen [38]. The first medication is CoQ10, which is involved in transporting electrons between complex 1 and 3 of the mitochondrial respiratory chain and helps generate ATP more efficiently [39,40]. Creatine generates ATP through the creatine phosphate shuttle and discourages the replication of abnormal mitochondria [41]. Carnitine brings fatty acids into the mitochondria so they can undergo beta oxidation to generate NADH [26,42]. Folic acid is a co-factor for several respiratory chain enzymes [43]. N-acetyl cysteine is a potent antioxidant [44,45], and the amino acid glutamine acts as an alternative energy source [46,47]. The doses of these medications vary for individual patients, but all patients have noted some benefit from them with regard to fatigue, exercise tolerance, and decreasing infection rate. With regard to glycogen storage diseases, patients are taught to avoid complex carbohydrates and supplement with simple sugars [48,49,50,51,52,53]. At the same time, since glycogen storage diseases are generally associated with mitochondrial dysfunction, we usually add the medications listed above that are used to treat mitochondrial diseases to patients’ treatment plan [54,55]. These patients saw significant improvements in fatigue and exercise tolerance with this regimen.

## 4. Discussion

We have demonstrated in this study that Sjogren’s patients with symptoms consistent with a metabolic disorder—fatigue, exercise intolerance, gastrointestinal dysmotility, and recurrent infections—often have mutations in genes important for metabolism. The most common gene mutations were observed in mitochondrial respiratory chain genes, although mutations in other mitochondrial genes and in genes involved in glycogen storage diseases were also observed. Previous studies have identified chromosomal aneuploidy in patients with autoimmune diseases [56,57]. Whether or not this contributes to the acquisition of metabolic abnormalities should be studied in the future. The identification of the metabolic disorder was helpful in suggesting therapies to improve disease symptoms. Patients saw symptomatic benefit from this treatment. No significant side effects were seen from the treatment.

Mitochondrial dysfunction has been observed in patients with Sjogren’s disease by several investigators [16,17,21,22,27,58,59]. However, no other investigators have treated Sjogren’s patients with the medications described in this manuscript for their metabolic disorders. No previous studies have described why knowing that Sjogren’s patients have metabolism disorders is clinically important.

The fact that so many patients with Sjogren’s disease have mitochondrial disorders raises the question as to whether mitochondrial dysfunction occurs secondary to inflammation in the salivary glands or whether it is a primary process contributing to the development of the disease. One way that mitochondrial dysfunction could contribute to disease pathogenesis is by decreasing the patient’s ability to handle infections, thus leading to more tissue damage and an increased likelihood that normal autorecognition is turned into pathologic autoreactivity [60,61,62,63]. Mitochondrial dysfunction could lead to the modification of various proteins and other molecules involved in signaling and genetic function [63]. Alternatively, inefficient mitochondrial function could lead to reliance on glycolytic metabolism, which tends to encourage the actions of the effector rather than regulatory lymphocytes and other immune cells [64,65,66,67,68,69,70,71]. Interestingly, IL-14 (a-taxilin) was recently shown to stimulate glycolysis [72]. The Il-14 transgenic mouse has been shown to be an excellent model for Sjogren’s disease [73,74]. Recent studies have demonstrated that blocking glycolysis inhibits the development of Sjogren’s disease manifestations in this animal model [75].

This manuscript has weaknesses because it describes patients followed as part of normal clinical service and does not describe a research study designed to address a particular research question. Furthermore, the patients were identified in a clinic that specializes in autoimmune disease and metabolic disorders, so it may not represent the types of patients with Sjogren’s disease seen in a general rheumatology or ophthalmology practice. All of the patients had autoantibodies associated with Sjogren’s disease, but only a few patients had SSA antibodies, which are the only autoantibodies in the official America–European diagnostic criteria for Sjogren’s disease [76]. Nonetheless, these patients all met the necessary clinical criteria and demonstrated autoreactivity through the presence of autoantibodies directed towards salivary and lacrimal gland antigens; the diagnostic criteria may have to be expanded to include additional autoantibodies. Furthermore, the expression of SSA versus SP1/CA6/ PSP may denote different stages of disease and/or different types of Sjogren’s disease that are driven by different metabolic and immunologic abnormalities [34,73,75,77,78,79].

Whether metabolic abnormalities are a primary or secondary function in patients with Sjogren’s disease, treatment based on these abnormalities is helpful for the patients symptomatically and may lead to other new forms of therapy.

## Figures and Tables

**Figure 1 ijms-26-04668-f001:**
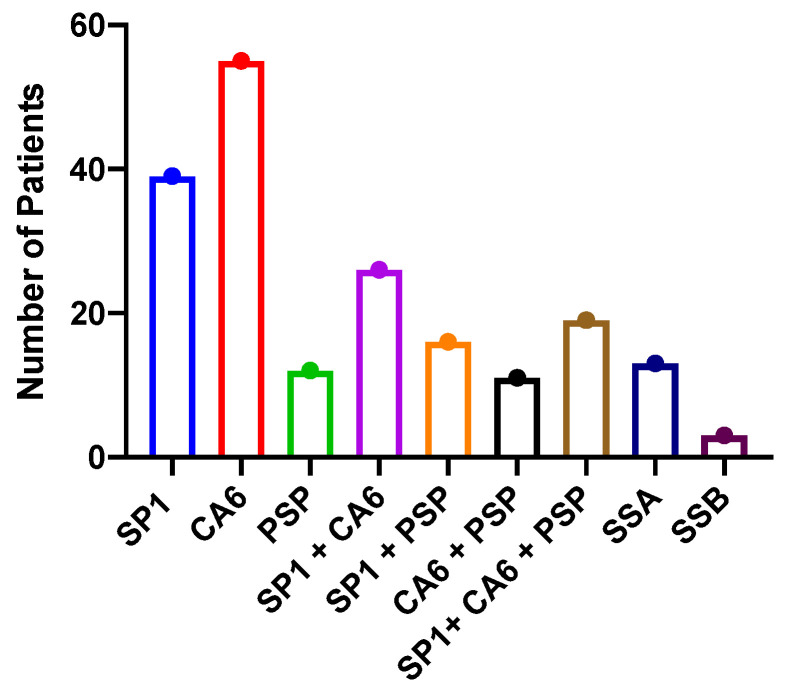
Autoantibodies identified in patients in this study. This figure demonstrates the number of patients with particular autoantibodies included in this study. SP1 = salivary protein 1; CA6 = carbonic anhydrase 6; PSP = parotid secretory protein; SS = Sjogren’s syndrome.

**Figure 2 ijms-26-04668-f002:**
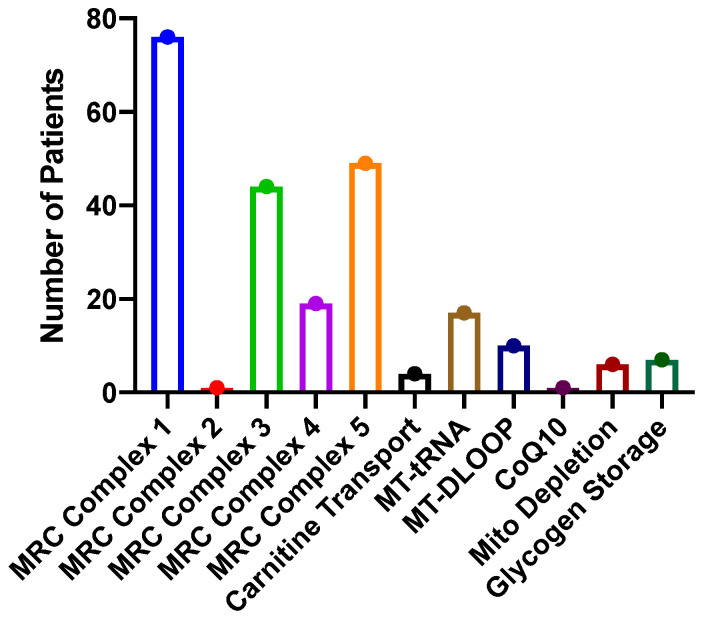
Mutations in genes involved in metabolism in patients with Sjogren’s disease. This figure demonstrates the number of patients with particular mutations in genes involved in metabolism. MRC = mitochondrial respiratory chain. MT = mitochondrial. Mitochondrial depletion genes included POLG, MCME, RRMP8, and thymidine kinase. Glycogen storage diseases included Pompe, Forber–Cori, McArdle’s, phosphofructokinase deficiency, and lactate dehydrogenase deficiency.

## Data Availability

Data are contained within the article.

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
