# Peer review of "Cellular Metabolic Disorders in a Cohort of Patients with Sjogren’s Disease"

_ijms, 2025, doi:10.3390/ijms26104668_

Round 1

Reviewer 1 Report

Comments and Suggestions for Authors

Dear Authors, your manuscript could be improved substantially. Why did you sent this document to case series section? I do not fully understand the aim of your work. Introduction section needs to be improved with more information about metabolic disorders. There are many papers in which you can support the background of this research and maybe focusing in genes related with metabolic disorders (please specify). In results, this section is poorly described. more information about metabolic disorders (clinical), more information about SjS; association between genes and AAs; a treatment pharagraph about metabolic disorders was included but it is not clear for me. Which is the relationship with the manuscript and the results section (???). Discussion section could be substantially improved, focusing in your results and contrasting with those previously published elsewhere. M and M, be more descriptive, not just enunciative.

Regards.

Author Response

RESPONSE TO REVIEWER:

Dear Authors, your manuscript could be improved substantially.

Why did you sent this document to case series section?

This manuscript was sent to the Case Series section because it describes a series of cases rather than asking a particular research question.

 I do not fully understand the aim of your work.

The purpose of this manuscript is to point out that many patients with Sjogren’s disease and many of the symptoms of Sjogren’s disease are related to an underlying metabolic disorder. This has never been documented before.

Introduction section needs to be improved with more information about metabolic disorders. There are many papers in which you can support the background of this research and maybe focusing in genes related with metabolic disorders (please specify).

The Introduction has been rewritten:

Understanding of the pathophysiology of autoimmune diseases has been rapidly advancing in the last several years. While dysregulated immune function has been appreciated for decades, appreciation of dysregulated metabolism in autoimmune diseases is relatively recent (1-7). Abnormal mitochondrial function was first observed in SLE (8-15) but has been observed in Sjogren’s disease as well (16-26).  Analysis of metabolic disorders in innate and adaptive immune cells of 30 patients with Sjogren’s disease documented alterations in multiple genes involved with mitochondrial metabolic pathways along with histologic abnormalities in mitochondria (17).  Abnormal mitochondria in salivary glands of patients with Sjogren’s disease have been observed in other studies as well (27, 28). One study suggested that abnormal production of cytochrome c in the salivary glands or patients with Sjogren’s disease contributed to apoptosis of salivary gland tissues (29). No previous studies have examined genes involved with systemic metabolic disorders in patients with Sjogren’s disease that could theoretically contribute to the development and / or manifestations of the disease. Fatigue, exercise intolerance, gastrointestinal dysmotility, accelerated osteoarthritis and difficulty handling infections is frequently seen in patients with adult-onset metabolic diseases (26, 30-32).  Many patients with Sjogren’s disease have similar symptoms (33). We observed in a limited numbers of patients previously that many of the symptoms attributable to Sjogren’s disease were in fact related to the underlying metabolic disease (24). We therefore sought to determine the number of patients with mutations in genes associated with metabolism in a cohort of patients with Sjogren’s disease seen in our clinics.  

 In results, this section is poorly described.

The results section provides the data on the patients. What autoantibodies they expressed and what mutations were identified in metabolic disease. A description of the clinical symptoms of each patient and what mutations they had is beyond the scope of this manuscript.

more information about metabolic disorders (clinical), more information about SjS; association between genes and AAs;

This is included in the introduction and discussion.

a treatment pharagraph about metabolic disorders was included but it is not clear for me.

This paragraph was included because we are the only group using this treatment regimen and the readers need to know what is meant by treatment of the metabolic disorders:

All patients received treatment for their underlying metabolic disorder. The treatment of mitochondrial disorders involves several medications, each of which works by a different mechanism, so a synergistic effect is seen (38). The first medication is CoQ10, which is involved with transporting electrons between complex 1 and 3 of the mitochondrial respiratory chain and helps generate ATP more efficiently (39, 40). Creatine  generates ATP through the creatine phosphate shuttle and discourages replication of abnormal mitochondria (41). Carnitine brings fatty acids into the mitochondria so they can undergo beta oxidation to generate NADH (26, 42). Folic acid is a co-factor for several respiratory chain enzymes (43). N-acetyl cysteine is a potent antioxidant (44, 45) and the amino acid glutamine  acts as an alternative energy source (46, 47). The doses of these medications vary for individual patients, but all patients have noted some benefit from them with regards to fatigue, exercise tolerance and decreasing infection rate.  With regards to the glycogen storage diseases, patients are taught to avoid complex carbohydrates and supplement with simple sugars (48-53). At the same time, since glycogen storage diseases are generally associated with mitochondrial dysfunction, we usually add the medications listed above that are used to treat mitochondrial diseases (54, 55). These patients saw significant improvement in fatigue and exercise tolerance with this regimen.

Which is the relationship with the manuscript and the results section (???).

The results section documents the fact that the patient had Sjogren’s disease and lists the mutations in metabolic genes that were identified in these patients

Discussion section could be substantially improved, focusing in your results and contrasting with those previously published elsewhere. M and M, be more descriptive, not just enunciative.

We have re-written the discussion as to the suggestion of the reviewer:

We have demonstrated in this study that Sjogren’s patients with symptoms consistent with a metabolic disorder, fatigue, exercise intolerance, gastrointestinal dysmotility and recurrent infections, often have mutations in genes important for metabolism. The most common gene mutations were in mitochondrial respiratory chain genes, although mutations in other mitochondrial genes and in genes involved with glycogen storage diseases were also observed.  The identification of the metabolic disorder was helpful in suggesting therapies to improve disease symptoms. Patients saw symptomatic benefit from this treatment.

                  Mitochondrial dysfunction has been observed in patients with Sjogren’s disease by several investigators (16, 17, 21, 22, 56-58). However, no other investigators have treated Sjogren’s patients with the medications described in this manuscript for their metabolic disorders.  No previous studies have described why knowing that Sjogren’s patients have disorders in metabolism is clinically important. 

The fact that so many patients with Sjogren’s disease have mitochondrial disorders raises the question as to whether mitochondrial dysfunction occurs secondary to the inflammation in the salivary glands or whether it is a primary process contributing to the development of the disease. One way that mitochondrial dysfunction could contribute to disease pathogenesis is by decreasing the ability to handle infections thus leading to more tissue damage and the increased likelihood that normal autorecognition is turned into pathologic autoreactivity (59-62). Mitochondrial dysfunction could lead to modification of various proteins and other molecules involved with signaling and genetic function (62). Alternatively, inefficient mitochondrial function could lead to reliance on glycolytic metabolism, which tends to encourage the actions of effector rather than regulatory lymphocytes and other immune cells (63-70). Interesting, IL-14 (a-taxilin) was recently shown to stimulate glycolysis (71). The Il-14 transgenic mouse has been shown to be an excellent model for Sjogren’s disease (72, 73). Recent studies have demonstrated that blocking glycolysis inhibits the development of Sjogren’s  disease manifestations in this animal model (74).

                  This manuscript has weaknesses because it describes patients followed as part of normal clinical service and does not describe a research study designed to address a particular research question. Furthermore, the patients were identified in a clinic that specializes in autoimmune disease and metabolic disorders, so it may not represent the types of patients with Sjogren’s disease seen in a general rheumatology or ophthalmology practice. All of the patients had autoantibodies associated with Sjogren’s disease, but only a few patients had SSA antibodies, which are the only autoantibodies in the official America – European diagnostic criteria for Sjogren’s disease (75). Nonetheless, these patients all met the necessary clinical criteria and demonstrated autoreactivity by the presence of autoantibodies directed towards salivary and lacrimal gland antigens – the diagnostic criteria may have to expand to include additional autoantibodies. Furthermore, expression of SSA versus SP1/CA6/ PSP may denote different stages of disease and/or different types of Sjogren’s disease that are driven by different metabolic and immunologic abnormalities (34, 72, 74, 76-78).

                  Whether metabolic abnormalities are a primary or secondary function in patients with Sjogren’s disease, treatment based on these abnormalities is helpful for the patients symptomatically and may lead to other new forms of therapy.

Reviewer 2 Report

Comments and Suggestions for Authors

The study of Julian Ambrus et al.investigates the presence of metabolic gene mutations in 194 patients with Sjogren’s disease and finds that 192 of them had mutations affecting mitochondrial function. Many symptoms commonly attributed to Sjogren’s, such as fatigue, gastrointestinal issues, and infections, were found to be related to these metabolic disorders. Treatment targeting the underlying metabolic abnormalities led to significant symptom improvement. These findings indicate that metabolic dysfunction may play a pivotal role in the onset or progression of Sjögren’s disease and should be taken into account when considering treatment strategies.

About the manuscript:

Were there any exclusion criteria of patients?

Did the authors perform any correlation analysis between specific mutations and the severity or type of symptoms?

Did the authors observe any relationship between specific autoantibody profiles (e.g., SP1 vs SSA) and the type of metabolic mutation?

Were any adverse effects observed during the treatment with mitochondrial-targeted therapies or dietary interventions?

Given that 89% of the study participants were female and 11% males, did the researchers observe any sex-specific patterns in the prevalence or type of metabolic mutations, symptom severity, or therapeutic outcomes in patients with Sjogren’s disease?

Exploring the relationship between genetic aneuploidy and autoimmunity may enhance the molecular diagnostic framework proposed in this study. I encourage the authors to comment on this possibility, ideally referencing recent literature.Specifically, I suggest including a reference on chromosomal aneuploidy and autoimmunity, such as the 2021 publication in Clinical and Experimental Immunology, to provide a more comprehensive context for the genetic findings discussed.

Author Response

Please see attached response. 

Round 2

Reviewer 1 Report

Comments and Suggestions for Authors

Dear SIRS, thank you for this new version of your manuscript. In order to be more clear about the aim of this case report (cases), I am understanding that metabolic disorders associated wth SjS are not conventional metabolic diseases (in example diabetes) you mean to mitochondrial disorders and genes related with mitochondrial metabolic pathways. That is the reason you are reporting genes mutations associated with metabolic  function. In that case I would like to suggest (with my respect) a change in manuscript´s title. In example "Metaboic disorders and gene mitochondrial mutations in a cohort of patients  with SjS" Think about it. Regards,

Author Response

Please see attached response. Thank you!

Reviewer 2 Report

Comments and Suggestions for Authors

the manuscript is ready for publication

Author Response

Thank you for your comments!